# Understanding patterns of loneliness in older long-term care users using natural language processing with free text case notes

**Sam Rickman**[ID]*, **Jose-Luis Fernandez**[ID], **Juliette Malley**[ID]

Care Policy and Evaluation Centre, The London School of Economics and Political Science, London, United Kingdom

* s.w.rickman@lse.ac.uk

## Abstract

Loneliness and social isolation are distressing for individuals and predictors of mortality, yet data on their impact on publicly funded long-term care is limited. Using recent advances in natural language processing (NLP), we analysed pseudonymised administrative records containing 1.1 million free-text case notes about 3,046 older adults recorded in a London council between 2008 and 2020. We applied three NLP methods—document-term matrices, pre-trained embeddings, and transformer-based models—to identify loneliness or social isolation. The best-performing model, a bidirectional transformer, achieved an $F_1$ score of 0.92 on a test set of unseen sentences. Using this model, we generated predictions for the full dataset and assessed construct validity through comparison with survey data and the literature. Our measure is associated with expected characteristics, such as living alone and impaired memory, and is a strong predictor of social inclusion services. Approximately 43% of individuals had a sentence indicating loneliness or isolation in their case notes at their initial care assessment, comparable to survey-based estimates. Unlike surveys, our indicator is linked to other administrative data, enabling development of models of service use with loneliness or isolation as independent variables. An open-source version of the model is available in a GitHub repository.

## Introduction

In 2021, public expenditure on long-term care was 1.98% of GDP in OECD countries [1]. In England, where the term adult social care is used to describe long-term support to complete activities of daily living, public spending was £23.7 billion (USD $30.4 billion) in 2022/2023 [2]. By 2038, projections indicate a 55% increase from 2018 levels in the number of older people receiving care, with public expenditure approximately doubling [3]. As most participants in national surveys do not receive publicly funded care [4,5], administrative care records provide a rich alternative about people using the social care system. In the UK, long-term care needs have been widely recorded in electronic databases since the 1990s [6],

**Data availability statement:** The full minimal dataset required to replicate the study findings is publicly available in the following GitHub repository: [https://github.com/samrickman/lonelinessmodel]. This has also been permanently archived with a DOI: [https://doi.org/10.5281/zenodo.14697740]. The code is licensed under the GNU General Public License v3.0. Instructions for running the

code, dependencies, and versioning information are provided in the repository to ensure reproducibility. There are no legal or ethical restrictions on sharing the code. This study uses pseudonymised, individual-level administrative social care records. Identifiable personal information was pseudonymised before egress. This includes names, addresses, email, telephone numbers, unique identifiers (e.g. NHS number), financial information, dates, and location details. Due to the sensitive nature of these records, we are unable to share the data publicly. Under the UK General Data Protection Regulation (GDPR), we are designated as data processors, and the data controller (a UK local authority) has not provided consent for public sharing of the data. We have agreed with the data controller not to disclose their identity to avoid increasing the probability of an individual's records being re-identified. The data used in this study cannot be shared publicly due to these legal and ethical restrictions. Requests for access to the underlying dataset would need to be directed to the Data Controller. To protect the anonymity of the local authority and individuals in the study, interested parties should contact cpec@lse.ac.uk, and the project support team can forward any enquiries appropriately.

**Funding:** This paper is based on independent research funded through the NIHR Policy Research Unit in Adult Social Care, reference PR-PRU-1217-21101. Funding was also received from the UK National Institute of Health and Care Research (NIHR) Applied Research Collaboration (ARC) North Thames under grant number NIHR200163. An additional grant was received from the NHS Digital Social Care Pathfinders initiative under the contract 8717. The views expressed are those of the authors and not necessarily those of the NIHR, ARC, NHS, or the Department of Health and Social Care. The funders had no role in study design, data collection and analysis, decision to publish, or preparation of the manuscript. The authors have declared that no competing interests exist.

**Competing interests:** The authors have declared that no competing interests exist.

and similar systems exist internationally [7,8]. Recent papers use natural language processing to extract information from free text electronic health records [9–14]. Few studies apply these methods to social care records [15–17], and none have focused on loneliness or social isolation.

This paper extracts an indicator of loneliness or social isolation from free-text administrative records. Needs assessment forms often lack structured indicators but include free text on social needs. Classified free-text data can be used to model care expenditure or service use data, such as care home entry, which are part of these records.

### The impact of loneliness and social isolation

Loneliness and social isolation are as significant predictors of mortality as smoking, obesity, or hypertension [18–20]. Social isolation is an "objective lack of relationships" [21], whereas loneliness is a "subjective, distressing feeling" when social relationships are inadequate [22].

Loneliness has been a longstanding priority for the WHO and governments, with its importance increasingly emphasised in recent years, particularly following the Covid-19 pandemic [23–28]. Yet, internationally, in recent decades, many countries have had a retrenchment in community care services towards personal care, with social support reduced [29–32]. Evidence suggests that loneliness increases long-term care use [33–37]. However, surveys often include few publicly funded care users [38–40], and records cannot be linked to detailed service use information, limiting its insights compared to administrative data.

Administrative records in England record eligibility-related social needs in structured and free text formats [41]. Distinguishing loneliness from social isolation in free text is challenging, as terminology often diverges from literature definitions. For example, "feels isolated" might refer to subjective loneliness or limited social contact. A 2024 paper by Patra et al. distinguished social support needs from psychiatric records, noting greater consistency and detail than typical social care notes [14]. Given the inconsistency in our dataset, we analyse loneliness and social isolation jointly. This combined approach is supported by findings showing both loneliness and isolation adversely affect older adults' mortality and is common in public health and work extracting Social Determinants of Health (SDoH) from clinical notes [12,18–20,42–44].

## Materials and methods

### Data collection

In England, every person requesting publicly funded care must receive an assessment under the Care Act 2014. In this paper we attempt to identify social isolation or loneliness from the free text notes of a London borough. Adult social care records are written by individuals employed by a local authority to assess needs and commission care. This generally consists of social workers, occupational therapists or care managers. Workers complete an assessment form, which is a snapshot of needs at a certain time containing both structured data and free text. Recording systems also contain case notes, which are free text fields to record ongoing work on the case over time. In Fig 1 we show how the assessment form and case notes appear to caseworkers.

### Ethics statement

This study uses secondary data from pseudonymised administrative social care records. We sought and were granted departmental ethics approval for the project on 30th May 2019 at the

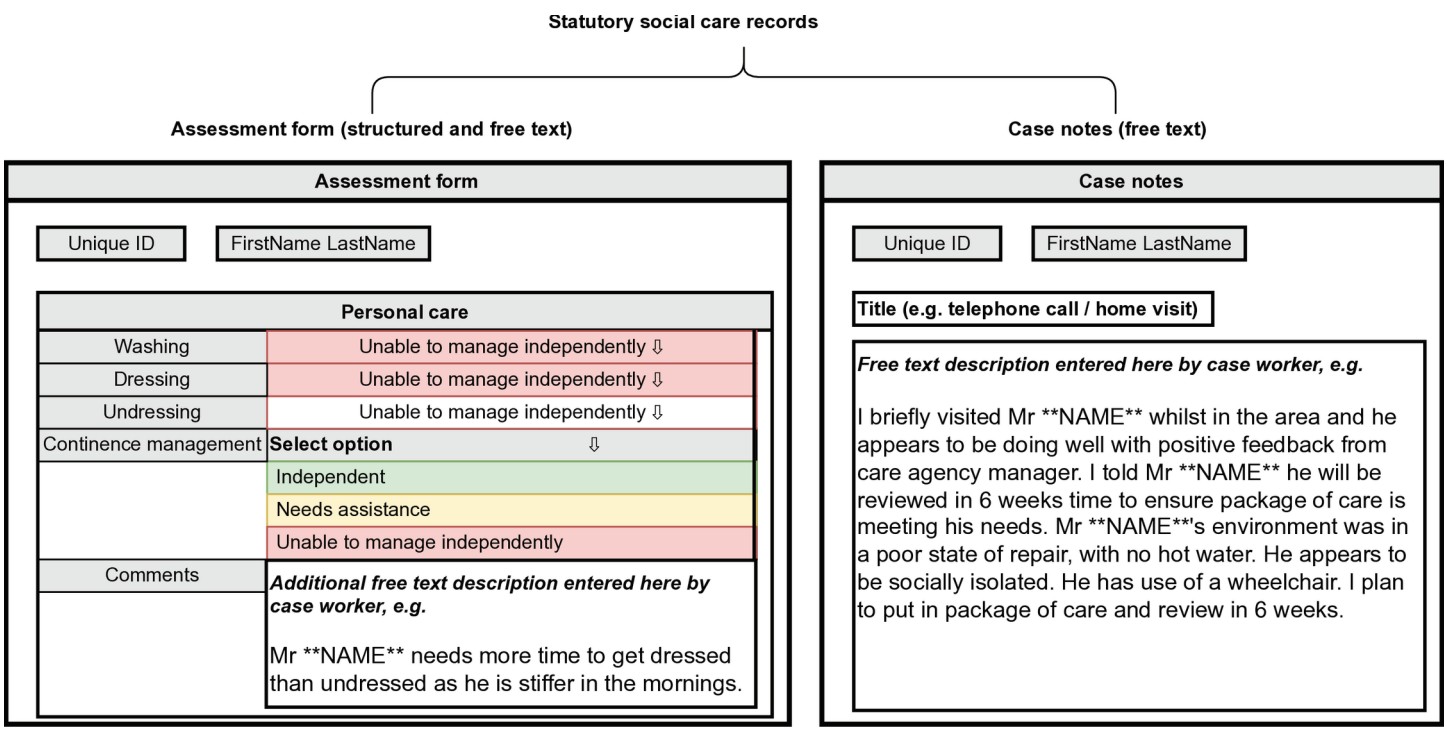

**Fig 1. Example of format of structured and unstructured data.**

London School of Economics and Political Science (LSE), in line with LSE's Research Ethics Policy and Procedure.

The data were pseudonymised prior to processing, including the removal or replacement of identifiable personal information such as names, addresses, email addresses, telephone numbers, unique identifiers (e.g., NHS numbers), financial information, and location details. A Data Processing Impact Assessment (DPIA) was carried out to ensure the protection of individuals' data privacy, and no automated decision-making processes were involved. The Data Flow Diagram is set out in S1 Fig in the Data Flow Appendix.

Details of the project were made available in the local authority's Privacy Notice and on a separate website informing individuals of and explaining the study, allowing individuals to opt out if desired. Individual consent for data use was not required, as the data were processed in line with the UK General Data Protection Regulation (GDPR) under the legal basis of legitimate interests. This legal basis allows processing of pseudonymised data for research purposes where it serves a social or public interest and individuals are informed and able to opt out. Permission for data processing was granted by the National Health Service NHS Confidentiality Advisory Group (CAG) in June 2020 (reference number 20/CAG/0043), which was renewed annually. CAG ensures that data processing complies with national regulations for handling confidential patient information in the UK.

### Data extraction and characteristics

A query was written to identify all individuals aged 65 or over on August 1st 2020 who had been receiving services for at least one year since 1st January 2016. Administrative records for these individuals were then extracted from the local authority database. Identifiable free

text data tokens were masked using the open-source text pseudonymisation software *PSCleaner* [45]. The data was then sent to an NHS Commissioning Support Unit, where identifiable structured data such as NHS numbers were removed. Finally, the data was transferred securely to the research team at the Care Policy and Evaluation Centre (CPEC) at the London School of Economics and Political Science (LSE).

The data includes all free text case notes recorded for individuals in the cohort between 2008 and 2020, as well as needs assessment and service receipt data. During this period, there were 3,046 individuals aged over 65 receiving long-term care. The data contains 10,821 assessment forms comprising 19.1 million words of free text, and 1.14 million case notes, containing 87.8 million words of free text. Case notes in the dataset encompass a wide range of updates related to the care and support of individuals. These include records of emails and telephone calls, descriptions of home visits, case screening and allocation, managerial direction, case summaries, allegations of abuse or neglect, as well as referrals to services such as occupational therapy, physiotherapy, and intermediate care. The volume of notes reflects the comprehensive documentation required in social care to capture various interactions and decisions throughout the course of care. There is significant variation in the distribution, with for example the top 50 individuals having 8 million words recorded (7.8% of the total), the same amount as the 850 individuals with the fewest. Summary statistics per person are presented in Table 1.

In addition to the free text case notes, structured data fields are routinely collected during the assessment process. These fields capture key demographic and personal information that is relevant for care planning and service provision. Structured data includes information such as gender, ethnicity, age, functional ability with activities of daily living (ADLs), and whether the individual lives alone. This information is collected directly by social care professionals during initial assessments and periodic reviews as part of standard care practices. These structured data fields provide important context for understanding the care needs of individuals and were used in conjunction with the free text data in our analysis. Of the 3,046 individuals, 61.2% were women, 47.8% White British, with a median age in 2020 of 81, and median of 3 years and 6 months of services received. These characteristics are set out in Table 2.

## Overview of model development and evaluation

This section outlines methods for model development and evaluation. We describe data preprocessing, manual text classification, and training machine learning algorithms. Model evaluation involves assessing performance metrics on a test set and examining construct validity by testing expected relationships with needs, demographics, and service use.

## Model development

We endeavoured to use as parsimonious a model as possible, beginning with count-based vector representations of words such as document-term matrices [46] and Term Frequency Inverse Document Frequency (Tf-idf) [47]. We also used the SpaCy large pre-trained word embeddings [48], and transformer-based representations, specifically RoBERTa and Distil-RoBERTa [49]. The overall process for training and comparing these models is set out in Fig 2. Unless otherwise stated, we used Python 3.9.7 in all the analysis [50].

For all approaches, we replaced the pseudonymised masks (e.g., ∗∗**NAME**∗∗, ∗∗**LOCATION**∗∗, which had been used to mask identifiable information) with randomly generated names and locations to ensure that the language models could correctly tokenise and parse the sentences. Retaining the pseudonymisation masks could have led to issues with tokenisation, as the

**Table 1. Quantity of free text per person.**

|  | Case notes | | Assessment | | Total |
|---|---|---|---|---|---|
|  | #notes | #words | #text fields | #words | #words |
| Mean | 377 | 28,850 | 148 | 6,740 | 35,115 |
| SD | 298 | 27,068 | 112 | 5,603 | 29,670 |
| Median | 302 | 21,444 | 123 | 5,212 | 27,330 |
| Min | 2 | 6 | 1 | 4 | 6 |
| Max | 2,585 | 407,283 | 869 | 44,196 | 408,404 |

**Table 2. Characteristics of individuals in the training and test set.**

| Group | N | Sentences (total) | Sentences (median) | Sentences (classified) | Min date | Max date | Service length | F (%) | WB (%) | YOB | Deceased |
|---|---|---|---|---|---|---|---|---|---|---|---|
| Train | 200 | 37.57263 | 1286 | 10083 | 2011-04-01 | 2022-04-15 | 3.03 | 62 | 47 | 1933 | 40 |
| Test | 200 | 305319 | 1238 | 3573 | 2011-04-01 | 2022-04-15 | 3.83 | 62 | 49 | 1934 | 43 |
| All | 3046 | 4807982 | 1289 | 13656 | 2010-01-29 | 2022-04-15 | 3.49 | 62 | 47 | 1934 | 42 |

**Notes:** F: Female. WB: White British. YOB: Median year of birth. Service length: Median time receiving statutory care services. Sentences (classified) is the number of sentences manually classified for model evaluation.

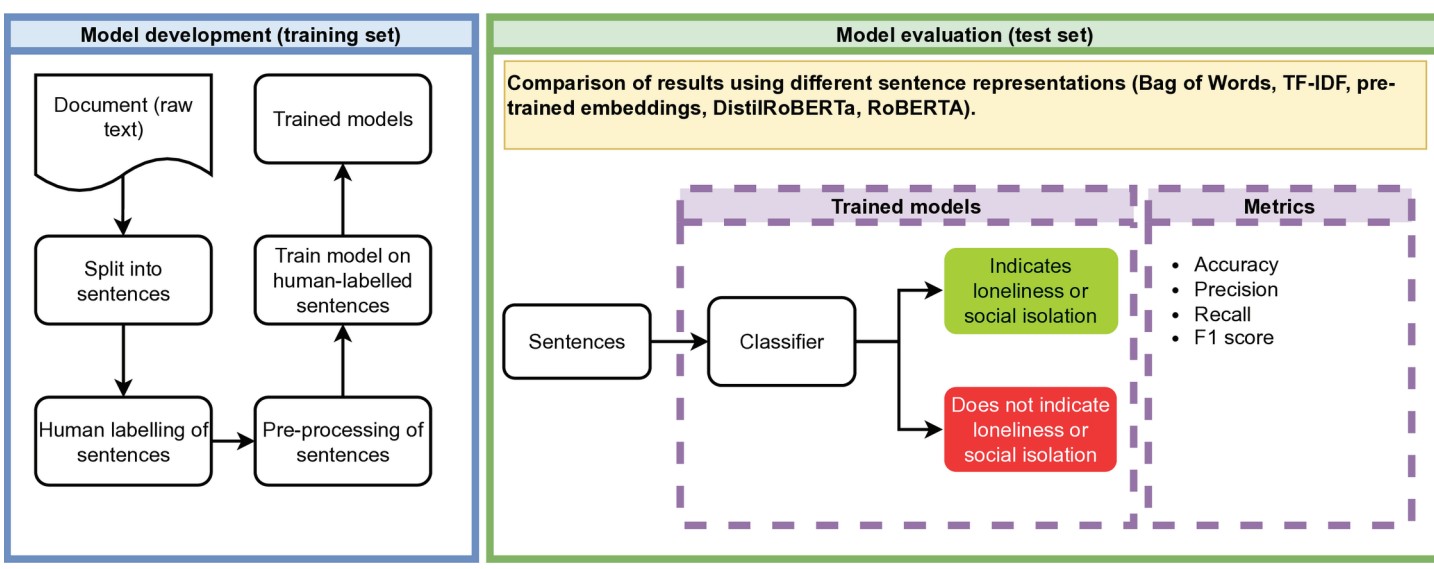

**Fig 2. Overall training and evaluation process.**

models may not have handled repeated placeholders effectively. For the count-based methods, we also lemmatised the text, converted it to lowercase, and removed stop words. We set out further details in Data pre-processing in S2 Fig in our Supporting Information document. We then divided the data into a training and test set, using stratified random sampling to ensure similar proportions of individuals in each set (see Table 2). Each set contained notes about 200 distinct individuals. We split each set by person to ensure that the test set did not contain sentences about individuals who are in the training set.

Human annotators manually classified 10,083 sentences in the training set and 3,573 sentences in the test set for model evaluation. These manually classified sentences represent a subset of the total number of sentences in the dataset, which exceeds 600,000. It would not

have been feasible to classify every sentence manually; instead, we focused on classifying sentences most likely to inform the development of the model. We defined a set of rules for annotators to determine which sentences to classify, using binary classification (either indicative or not indicative of loneliness or social isolation). These rules covered statements such as when a person explicitly expressed feeling lonely, had little social contact, or received referrals to services like befriending. Conversely, sentences indicating practical support needs, support for safety or cognition, or day centre attendance for carer respite were classified as not indicative of loneliness or social isolation. The full set of rules is detailed in S1 Text. Our interrater reliability measures produced Cohen's $\kappa$ [51] of 0.89 (95% CI 0.84–0.94) and Krippendorff's $\alpha$ [52] of 0.89 (95% CI 0.89–0.93). The maximum level of agreement in both cases is 1, and 0.89 represents excellent levels of agreement beyond chance [53,54]. The training dataset was imbalanced, with 9,383 sentences in the negative class (not indicative of loneliness or social isolation) and 700 in the positive class.

We implemented three approaches for the representation of words:

1. Count-based approaches: We split each sentence into lemmatised, word-level tokens. Each sentence was represented by a raw count of the number of times a word appears in it (a document-term matrix) [46]. We also applied Term Frequency Inverse Document Frequency (Tf-idf) [55] to transform the count matrix to a weighted representation, reducing the weighting of higher frequency words across all documents.

2. Pre-trained vectors: We used the Spacy large English model [48], which represents language through dense embeddings [56], where words which have similar semantic meanings are clustered together in vector space. We took the mean of each dimension to create a single 300-dimensional vector to represent each sentence.

3. Transformer-based approaches: We used the RoBERTa *base* model, which has 12 hidden layers, 768 dimensions and 12 heads [49]. This was relatively computationally expensive to fine-tune, so for comparison we also used DistilRoBERTa, which has identical parameters except it has 6 hidden layers, and is around twice as fast to train. In both cases, we used the HuggingFace implementation of each model's tokenizer to split each sentence into sub-word tokens [57,58].

We describe these approaches in more detail in S1 Text. After pre-processing, vectorising and labelling each sentence, the problem becomes a binary classification task. For both the count and pre-trained embedding based approaches, we evaluated five classification algorithms. We used $k$ fold cross-validation to avoid overfitting on the training set, choosing 5 folds for $k$ as a value which tends to elicit reasonably high accuracy [59] while reducing training time compared with higher values. We used five classification algorithms: class-weighted logistic regression, bootstrap aggregation, random forest, quadratic discriminant analysis and a feed-forward neural network. Again we set these out in the Classification algorithms section of S1 Text. For the transformers approach, the HuggingFace implementation of both the RoBERTa and DistilRoBERTa models contain a classification head. We trained this final layer of the model on the labelled, tokenised sentences using the HuggingFace Transformers and PyTorch libraries [60,61]. Our final model had a training batch size of 16 sentences, with 500 warm up steps, and weight decay of 0.01. The weight decay parameter is bounded between 0 and 1, with 0.01 indicating relatively low L2 regularisation, which can help the model fit the smaller, positive classes more accurately, but can risk overfitting. The final output layer produces a predicted probability for the negative and positive classes (either indicative or not indicative of loneliness or social isolation). During training the parameters of the classification head were optimised using binary cross-entropy loss, which measures the difference between the predicted probabilities and the true labels.

## Model evaluation

We evaluate the model's accuracy by comparing performance metrics (accuracy, precision, recall, $F_1$) on a test set of 3,573 unseen sentences drawn from individuals not included in the training data (3026 in the negative class and 547 in the positive class). We assess construct validity of the indicator derived from the best-performing classification model by analysing associations between the NLP model output and demographic characteristics, and comparing this with associations in survey data. We also conduct logistic regression to assess whether the model's loneliness or isolation predictions are associated with the use of services typically related to social support needs.

**Construct validity: Comparison with survey data.** Using the best-performing model, we assess construct validity by classifying free text from the initial assessments of 1,331 individuals at their first contact with statutory care services. We analyse text from assessment forms and case notes within 90 days of the assessment, with the indicator of loneliness or isolation treated as binary. We derive four metrics: individuals with no positive sentences (*Neither*), those with a positive sentence only in their assessment (*Assessment*), those with a positive sentence only in case notes (*Case notes*) and those with both (*Assessment*). This is set out in Table 3.

We then compare the results of the model predictions with pooled data from waves 6–9 (2012–2019) of ELSA [40]. This secondary dataset, collected as part of a large national survey through structured interviews and self-reported questionnaires, provides a validated source of information on the characteristics of older adults in England. We use ELSA data for all older adults who stated that they had care needs and received publicly funded care (N = 995 unique individuals with 1361 total observations over the period). We pool the results due to the low number of responses in some groups. We tabulate responses to the ELSA Center for Epidemiological Studies Depression Scale (CES-D) loneliness question [62]. We also compare our results to the three UCLA loneliness scale questions within ELSA, converting a total score of 6 or more into a binary indicator of loneliness, as in Hanratty et al. [33]. As our model measures loneliness and social isolation, in the ELSA data we also establish which individuals are socially isolated according to the Social Network Index (SNI) defined in Minicuci et al. (2016) [63].

We compare the results with ELSA graphically, by examining the proportion of people in our data and in ELSA who appear lonely or socially isolated, broken down by demographic characteristics and care needs. We also conduct a Pearson's $\chi^2$ test of independence [64] of each need or demographic factor with loneliness or isolation, to establish whether there are the same associations between our indicator of loneliness and those found in ELSA. Finally, we conduct a logistic regression of all these factors and loneliness or isolation, to establish which factors remain significant after controlling for characteristics such as living alone.

**Construct validity: Predicting service receipt for loneliness or isolation.** Our data includes information on whether individuals are attending day centres, which are community-based services provided for older people at risk of loneliness or social isolation [30]. To assess whether, as we expect, our indicator is associated with day centre attendance, we generated predictions of loneliness or social isolation using our best-performing model, the RoBERTa-based language model. Next, we conducted a logistic regression to examine whether these predictions were associated with the receipt of day centre services within 90 days of the initial assessment, for the 1,331 individuals whose initial assessment could be identified. To ensure that the RoBERTa model was not simply picking up cases with more notes, or cases driven by demographic characteristics rather than actual loneliness or isolation, we included the number of notes and relevant demographic variables as controls in

**Table 3. Classification outcomes across assessments and case notes.**

|  | Case notes | | |
|---|---|---|---|
|  |  | 0 | 1 |
| Assessment | 0 | Neither | Case notes |
|  | 1 | Assessment | Both |

the logistic regression model. This allowed us to verify that the RoBERTa model's predictions were not confounded by factors such as a greater volume of documentation or demographic differences, rather than genuine cases of loneliness or social isolation. The logistic regression model is specified in Eq (1).

$$\log \frac{p}{1-p} = \beta_0 + \beta_1 SIL + \beta_2 notes + \beta_3 sex + \beta_4 ethnicity + \beta_5 age +$$
$$\beta_6 pc + \beta_7 memory + \beta_8 safety + \beta_9 alone$$
(1)

Where $p$ is the probability of receiving day services in the first 90 days, $SIL$ is the binary prediction of social isolation or loneliness generated by our model, $notes$ is the number of sentences written within 90 days of assessment, $sex$ is a binary variable where 1 indicates male, $ethnicity$ is a binary indicator of white or non-white and $age$ is age of the person receiving care in years. Additionally, we include as continuous variables the following rank of severity of needs, where higher indicates more care needs. $pc$ is personal care needs (the sum of mobility, toileting and dressing), $memory$ is the score for memory and cognition, $safety$ is the extent to which the person is aware of their own safety and risk and $alone$ is a binary indicator of whether an individual lives alone. The demographic and needs-related scores are extracted from the structured data of the initial assessment.

## Results

We present a set of results for each method of model evaluation. Firstly, we evaluate the performance of each model against the test set. Secondly, we run the best-performing model on text recorded within 90 days of every initial assessment and compare the significance of association with demographic characteristics with survey data from ELSA. Finally, we present the logistic regression of the results of the best-performing model on day centre attendance.

### Model performance on the test set

In Table 4 we detail the accuracy, precision, recall and $F_1$ score [65] of each model on the test set of 3,573 labelled sentences not seen by the training set. The transformer-based models considerably outperform all other models, with DistilRoBERTa achieving an $F_1$ score of 0.86 and RoBERTa 0.92. The pre-trained Spacy embeddings outperformed all non-transformers based approaches when classes were predicted using a feed-forward neural network, with an $F_1$ score of 0.61. However, using the same embeddings, the neural network only slightly outperformed logistic regression, which had an $F_1$ score of 0.58. The count-based approaches were not effective at prediction using any of the classification methods. High accuracy alongside low precision, recall, and F1 scores in some models reflects the imbalanced dataset, as models like random forest almost exclusively predict the majority class, inflating accuracy while failing to classify minority cases. In Fig 3, we present a confusion matrix comparing the predictions of the best-performing model against the classes defined by human annotation.

**Table 4. Accuracy and F1 score of classification models.**

| Classifier | Accuracy | Precision | Recall | F1 |
|---|---|---|---|---|
| **Transformers** | | | | |
| RoBERTa | 0.97 | 0.95 | 0.87 | 0.92 |
| DistilRoBERTa | 0.96 | 0.90 | 0.82 | 0.86 |
| **Pre-trained embeddings** | | | | |
| Feed forward NN | 0.90 | 0.78 | 0.50 | 0.61 |
| Logistic regression | 0.83 | 0.46 | 0.77 | 0.58 |
| QDA | 0.84 | 0.45 | 0.28 | 0.34 |
| Bagging | 0.85 | 0.66 | 0.07 | 0.13 |
| Random forest | 0.85 | 1.00 | 0.02 | 0.03 |
| **Tf-idf** | | | | |
| QDA | 0.27 | 0.15 | 0.83 | 0.26 |
| Bagging | 0.82 | 0.15 | 0.03 | 0.06 |
| Feed forward NN | 0.84 | 0.22 | 0.01 | 0.03 |
| Logistic regression | 0.84 | 0.07 | 0.01 | 0.01 |
| Random forest | 0.84 | 0.00 | 0.00 | 0.00 |
| **Document-term matrix** | | | | |
| QDA | 0.23 | 0.15 | 0.83 | 0.25 |
| Bagging | 0.83 | 0.13 | 0.02 | 0.04 |
| Feed forward NN | 0.84 | 0.24 | 0.02 | 0.03 |
| Logistic regression | 0.84 | 0.06 | 0.00 | 0.00 |
| Random forest | 0.84 | 0.00 | 0.00 | 0.00 |

**Construct validity: Comparison with survey data.** The overall proportion of individuals with at least one case note indicating loneliness or social isolation according to our model is 0.44 (95% CI 0.42–0.47), and the proportion with at least one sentence indicating the same in their needs assessment is 0.43 (95% CI 0.40–0.45). This compares with a proportion of 0.38 in ELSA (95% CI 0.32–0.43) who are lonely according to the CES-D measure or SNI isolated, and 0.45 (95% CI 0.39–0.51) who are lonely according to the UCLA measure, or SNI isolated. The overall proportions are similar to the UCLA loneliness measure, and this holds for many characteristics. We present in Fig 4 a breakdown of these proportions by demographic and needs-related factors. While these similarities are reassuring, there are differences between the results of our model and ELSA. For example, the difference in loneliness between individuals who live alone and live with others is wider in ELSA than in our data. We present the results in Fig 4 in tabular form in the S1 Text document.

We set out in Table 5 the results of the $\chi^2$ test of independence between loneliness or isolation and the needs-related factors in both our results and ELSA. We also present the results of the combined indicators, *Either* and *Both* . The $\chi^2$ tests reveal both similarities and differences in the associations between our RoBERTa-based indicator and the ELSA measures of loneliness and social isolation (CES-D and UCLA combined with SNI). Both our indicator and the survey data show a strong association between loneliness and living alone. However, our indicator also identifies a significant link between memory issues and loneliness, which is not found in the ELSA data. Additionally, ELSA data shows that people receiving unpaid care are more likely to be lonely, a pattern not reflected in our findings.

We set out the results of the $\chi^2$ test and regression of the association with needs and demographic factors in Table 5. We assessed multicollinearity using the generalised variance inflation factor (GVIF), with a maximum value of 1.3, well below the typical threshold of 4–10 [66, 67]. The regression output indicates that in ELSA, living alone is by far the most significant predictor, though requiring support with shopping and presence of unpaid care are also significant. Across all four of our measures, living alone is also a very important predictor of

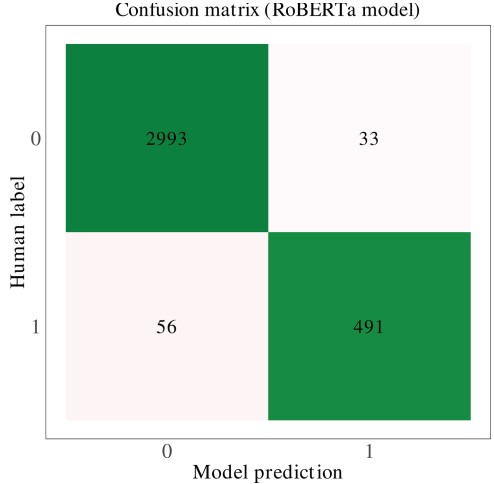

**Fig 3. Confusion matrix (RoBERTa).**

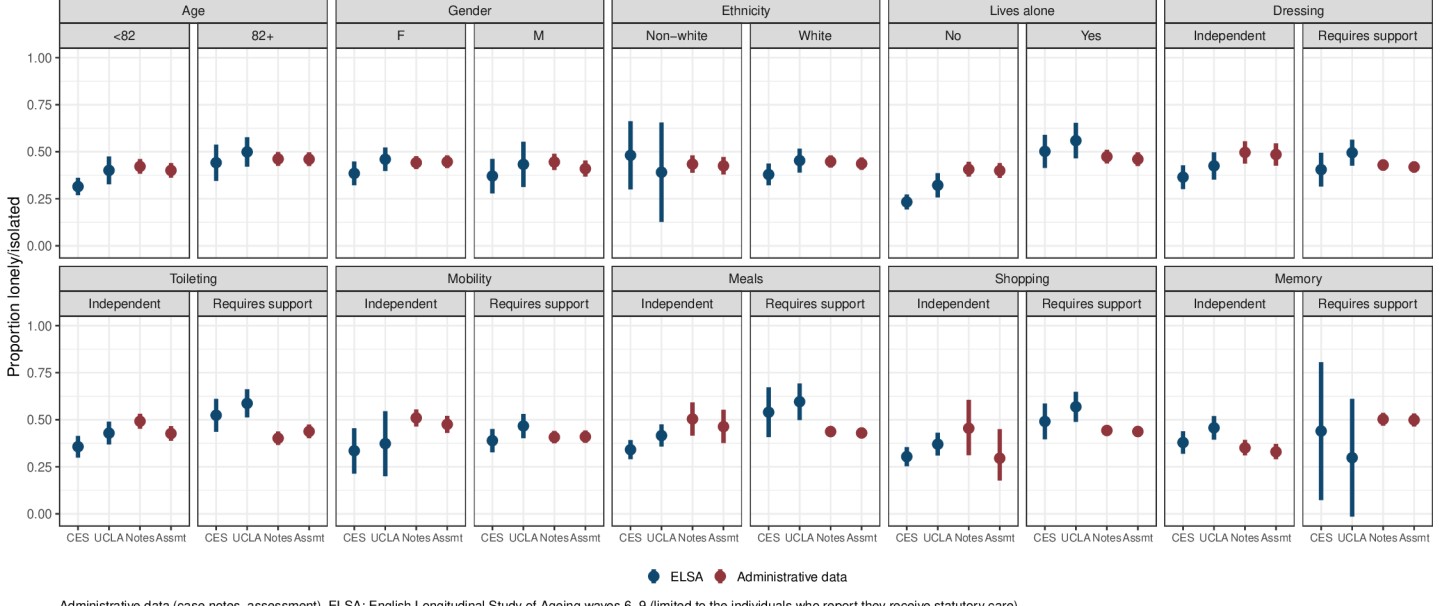

**Fig 4. Proportion of lonely/isolated by demographic characteristics: Administrative and survey data.**

loneliness or isolation. The coefficient is around the magnitude of that for memory, where individuals who have memory problems are more likely to be lonely or socially isolated.

This discrepancy between our results and ELSA may be due to differences in the cohorts or the nature of the data, as ELSA data is self-reported, while administrative assessments of functional ability are recorded by professionals. Although we have taken a subset of individuals from ELSA who are older people receiving local authority care, individuals in the administrative data have higher needs than those in ELSA (see Table 6). We do not consider this

**Table 5. Factors in structured data associated with loneliness and social isolation: Administrative data and ELSA.**

| | Administrative records | | | | ELSA | |
|---|---|---|---|---|---|---|
| | Assessment | Case notes | Either | Both | CES-D | UCLA |
| **Chi-sq test** | | | | | | |
| Dressing | 0.048* | 0.047* | 0.254 | 0.001*** | 0.398 | 0.099· |
| Ethnicity | 0.688 | 0.635 | 0.655 | 0.619 | 0.606 | 0.526 |
| Lives alone | 0.027* | 0.013* | 0.033* | 0.003** | <0.001*** | <0.001*** |
| Meals | 0.471 | 0.154 | 0.59 | 0.066· | <0.001*** | <0.001*** |
| Memory | <0.001*** | <0.001*** | <0.001*** | <0.001*** | 0.462 | 0.605 |
| Mobility | 0.02* | <0.001*** | 0.071· | <0.001*** | 0.509 | 0.016* |
| Safety & risk | 0.284 | 0.126 | 0.869 | 0.477 | 0.737 | 0.624 |
| Sex (F) | 0.188 | 0.896 | 0.814 | 0.278 | 0.262 | 0.181 |
| Shopping | 0.062· | 0.878 | 0.473 | 0.25 | <0.001*** | <0.001*** |
| Toileting | 0.68 | 0.001*** | 0.467 | 0.013* | 0.046* | 0.002** |
| Unpaid care | 0.726 | 0.258 | 0.689 | 0.657 | <0.001*** | <0.001*** |
| **Logistic regression** | | | | | | |
| Age | 1.01 (0.99–1.02) | 1.01 (1.00–1.03) | 1.01 (1.00–1.02) | 1.01 (1.00–1.03) | 1.01 (0.99–1.03) | 1.00 (0.99–1.02) |
| Dressing | 0.87 (0.78–0.97)** | 0.91 (0.82–1.01)· | 0.88 (0.79–0.99)* | 0.87 (0.78–0.97)* | 0.88 (0.62–1.23) | 0.92 (0.66–1.28) |
| Ethnicity | 0.96 (0.76–1.23) | 0.98 (0.77–1.24) | 0.99 (0.77–1.27) | 0.95 (0.72–1.23) | 1.60 (0.47–5.34) | 1.33 (0.36–5.05) |
| Lives alone | 1.37 (1.08–1.75)** | 1.38 (1.09–1.76)** | 1.52 (1.18–1.95)*** | 1.39 (1.06–1.82)* | 6.12 (4.36–8.70)*** | 3.57 (2.62–4.90)*** |
| Meals | 1.02 (0.90–1.16) | 1.09 (0.96–1.23) | 1.09 (0.95–1.24) | 1.04 (0.91–1.19) | 1.36 (0.89–2.07) | 1.31 (0.86-1.99) |
| Memory | 1.40 (1.25–1.57)*** | 1.40 (1.25–1.57)*** | 1.50 (1.33–1.70)*** | 1.47 (1.30–1.67)*** | 1.14 (0.47–2.71) | 0.65 (0.26–1.57) |
| Mobility | 0.89 (0.81–0.99)* | 0.99 (0.90–1.10) | 1.01 (0.90–1.13) | 0.85 (0.76–0.96)** | 0.86 (0.56–1.32) | 1.19 (0.79–1.80) |
| Safety & risk | 1.03 (0.92–1.15) | 0.94 (0.84–1.05) | 1.00 (0.89–1.12) | 0.96 (0.85–1.09) | 1.35 (0.55–3.26) | 1.17 (0.48–2.85) |
| Sex (F) | 1.15 (0.91–1.45) | 0.89 (0.70–1.12) | 0.97 (0.76–1.24) | 1.06 (0.82–1.37) | 0.92 (0.66–1.27) | 0.99 (0.72–1.35) |
| Shopping | 1.09 (0.93–1.28) | 1.02 (0.88–1.20) | 1.06 (0.90–1.24) | 1.07 (0.90–1.28) | 1.61 (1.13–2.29)** | 1.45 (1.03–2.05)* |
| Toileting | 1.04 (0.95–1.15) | 0.88 (0.80–0.97)* | 0.93 (0.84–1.03) | 0.97 (0.87–1.08) | 1.25 (0.79–1.97) | 1.46 (0.94–2.26)· |
| Unpaid care | 1.01 (0.78–1.30) | 0.88 (0.68–1.14) | 0.94 (0.71–1.23) | 0.92 (0.70–1.23) | 1.84 (1.08–3.19)* | 2.22 (1.27–3.98)** |

**Notes:** ELSA: English Longitudinal Study of Ageing. CES-D: Center for Epidemiological Studies Depression. Chi-sq results are p-values. Logistic regression results are coefficients (0.95 CI).
· $p < 0.1$
* $p < 0.05$
** $p < 0.01$
*** $p < 0.001$

**Table 6. Comparison of demographic and ADL needs between ELSA waves 6–9 and administrative data.**

| | Administrative | | ELSA | |
|---|---|---|---|---|
| | N (%) | N Unique | N (%) | N Unique |
| Ethnicity (non-white) | 364 (33%) | 364 | 18 (4%) | 15 |
| Toileting (requires support) | 570 (52%) | 570 | 160 (38%) | 132 |
| Lives alone | 608 (55%) | 608 | 255 (60%) | 203 |
| Memory (has needs) | 664 (60%) | 664 | 50 (12%) | 49 |
| Sex (F) | 686 (62%) | 686 | 267 (63%) | 208 |
| Awareness of risk (impaired) | 806 (73%) | 806 | 90 (21%) | 81 |
| Unpaid care (receives) | 819 (74%) | 819 | 344 (81%) | 277 |
| Dressing (requires support) | 877 (80%) | 877 | 293 (69%) | 233 |
| Meals (requires support) | 998 (91%) | 998 | 276 (65%) | 219 |
| Shopping (requires support) | 1066 (97%) | 1066 | 336 (79%) | 272 |

**Notes:** ELSA: English Longitudinal Study of Ageing waves 6–9 (limited to the subset of individuals who report they receive statutory care). N Unique: number of unique individuals (as data is pooled). Administrative values are recorded by care managers in structured data. ELSA values are from the variables: raracem, toilta, hhres, slfmem, sex, dangera, rcaany_e, dressing, mealsa, shopa.

a barrier to comparing the datasets, but we do consider it when interpreting the results. We elaborate on this in the Discussion section.

**Table 7. Logistic regression: Association of loneliness extracted from free text with services received for loneliness.**

| | Odds ratio (RoBERTa model) | | | |
|---|---|---|---|---|
| | Assessment | Both | Either | Notes |
| Lonely/isolated (prediction) | 5.74 (3.02–11.87, p < 0.001)*** | 8.35 (4.57–16.09, p < 0.001)*** | 9.65 (3.47–40.16, p < 0.001)*** | 8.11 (3.80–20.11, p < 0.001)*** |
| N notes | 1.00 (1.00–1.00, p = 0.319) | 1.00 (1.00–1.00, p = 0.080)˙ | 1.00 (1.00–1.00, p = 0.118) | 1.00 (1.00–1.00, p = 0.035)* |
| Sex: Male | 1.12 (0.64–1.94, p = 0.681) | 1.08 (0.61–1.90, p = 0.777) | 1.01 (0.58–1.74, p = 0.969) | 0.95 (0.54–1.65, p = 0.869) |
| Ethnicity: White | 1.21 (0.69–2.17, p = 0.516) | 1.24 (0.70–2.27, p = 0.464) | 1.21 (0.69–2.16, p = 0.516) | 1.23 (0.70–2.22, p = 0.476) |
| Age | 0.98 (0.95–1.02, p = 0.393) | 0.98 (0.95–1.02, p = 0.369) | 0.98 (0.95–1.02, p = 0.351) | 0.98 (0.95–1.02, p = 0.358) |
| Personal care score | 0.68 (0.53–0.85, p = 0.001)*** | 0.74 (0.58–0.94, p = 0.014)* | 0.66 (0.52–0.83, p = 0.001)*** | 0.69 (0.54–0.87, p = 0.002)** |
| Memory score | 1.82 (1.41–2.36, p < 0.001)*** | 1.77 (1.36–2.32, p < 0.001)*** | 1.83 (1.43–2.37, p < 0.001)*** | 1.83 (1.42–2.37, p < 0.001)*** |
| Safety & risk score | 0.99 (0.76–1.27, p = 0.911) | 1.00 (0.78–1.30, p = 0.978) | 1.00 (0.78–1.29, p = 0.993) | 1.02 (0.79–1.31, p = 0.893) |
| Lives alone | 0.33 (0.18–0.59, p < 0.001)*** | 0.33 (0.18–0.60, p < 0.001)*** | 0.34 (0.18–0.60, p < 0.001)*** | 0.34 (0.19–0.62, p = 0.001)*** |

˙ p < 0.1
* p < 0.05
** p < 0.01
*** p < 0.001

**Construct validity: Predicting service receipt for loneliness or isolation.** We include the results of the day centre services regression in Table 7. Accounting for the number of notes and demographic factors, the model output remains a strong predictor of whether an individual is in receipt of day centre services. The maximum GVIF for any indicator is less than 1.4.

## Discussion

The goal of this analysis was to extract an indicator of loneliness or social isolation from free text. Our key finding is that a RoBERTa-based transformer model can produce this indicator with high accuracy ($F_1$ = 0.92), outperforming simpler methods like document-term matrices or pre-trained embeddings. Transformer models handle the complexity of adult social care records better, likely due to their attention mechanism, which captures context-dependent distinctions. Example sentences in Fig 5 illustrate cases where transformer models succeed while other methods do not, reflecting their ability to process the complex, unstructured data that is found in adult social care records.

We validated the indicator by applying the model to initial assessments of 1,331 individuals and comparing its predictions to survey data and the literature. The indicator strongly predicts the receipt of social inclusion services and aligns with known associations, such as living alone. However, there are differences from survey findings: in ELSA, living alone shows a stronger link to loneliness, likely because marital status, a component of the ELSA SNI indicator [63], is strongly correlated with living alone. Conversely, our indicator identifies a significant association between loneliness and memory issues, which is absent in ELSA.

These discrepancies may stem from differences in datasets. Administrative records include higher-need individuals than ELSA, where only 12% report impaired memory compared to 60% in administrative data (Table 6). Survey attrition may exclude those with severe needs [68], while self-reports in ELSA could understate functional impairments due to social desirability bias or cognitive issues [69,70]. Prior research shows correlations between self-reported and actual ability can be as low as 0.2, with individuals often overstating their mobility [71,72].

Self-reports in ELSA may also explain other differences, such as the link between unpaid care and loneliness observed in ELSA but not in our data. Unpaid care in ELSA may act as a proxy for need, which may not be fully captured by ELSA's functional questions. Despite

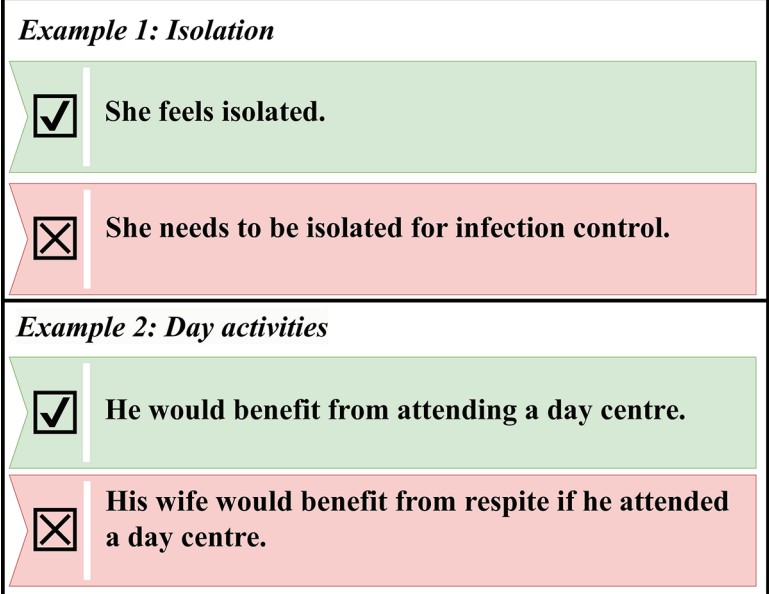

**Fig 5. Examples of polysemy in adult social care case notes.**

small sample sizes in ELSA, we retained all older individuals receiving publicly funded care but interpret these findings cautiously. While challenges exist in comparing self-reported data to social care records, our indicator aligns well with measures where self-reporting is less likely to differ from professional assessment, like gender, where both sources show slightly higher loneliness among women, reinforcing its validity.

In the administrative data, loneliness or isolation appears less common among individuals with higher physical care needs. While no single physical ADL shows a consistent negative association across all measures, there is a general trend that requiring more physical support correlates with reduced loneliness. This could reflect a real effect, as suggested by the negative (though not significant) dressing coefficient in ELSA, or it may result from how workers prioritise recorded needs. For individuals with high physical care needs, workers may focus on immediate risks, such as falls or pressure ulcers, rather than loneliness, limiting the classifier's ability to capture true prevalence. Additionally, unlike administrative data, ELSA shows no significant association between memory problems and loneliness, despite literature suggesting such a link [73,74].

The comparison with ELSA is challenging to interpret, owing to the apparent differences in the population and that both care needs and loneliness are self-reported in ELSA but not in administrative data. We are therefore reassured by the results in Table 7 of the probability of receipt of day centre services within 90 days of the first assessment. It is clear that the indicator of loneliness or isolation is a strong and significant predictor of whether an individual receives services for social inclusion. This holds when controlling for the number of notes and demographic factors, suggesting that our indicator is picking up a distinct phenomenon. It also leads to the reassuring conclusion that workers who record that a person is lonely or isolated are much likelier to put in place services for this need.

Interpreting the model's output involves determining which of the four metrics (*Assessment*, *Case notes*, *Either*, or *Both*) is most appropriate. All metrics are associated with similar demographic and needs-related factors, likely due to the binary nature of the

measurement. This binary approach oversimplifies loneliness, which varies in intensity, but this might be captured to an extent by combining the binary metrics. For instance, the *Assessment* and *Case notes* metrics (around 43% prevalence each) are approximately as prevalent as individuals who are SNI isolated or have a UCLA loneliness score of 6 or higher (45%). In contrast, the *Both* metric (26% prevalence) may identify individuals with more severe loneliness, comparable to those scoring 9 out of 9 on the UCLA scale (26%). The *Either* metric (62% prevalence) aligns with UCLA scores between 4 (71%) and 5 (58%). The choice of metric depends on the policy goal: a higher threshold might target those with the highest need, while a lower one could cast a wider net for preventive interventions. However, this is speculative, and whether these proportions are in fact indicative of intensity is an empirical question that requires further validation.

## Limitations

Our findings have some limitations. For pre-trained embeddings, we used mean pooling to represent sentences, but methods that summarise spans of embeddings may improve performance [75]. Similarly, count-based approaches could have benefited from using n-gram co-occurrence matrices to capture contextual relationships more effectively. These enhancements might have increased the $F_1$ score of simpler NLP methods. However, we ensured robust evaluation by testing a range of classifiers, including boosting, bagging, logistic regression, MLP, and random forests. Additionally, with transformer models, we achieved strong results using default parameters without hyperparameter tuning, suggesting potential for further optimisation in these approaches too.

Our binary classification of loneliness or social isolation oversimplifies the concept, as it cannot capture variations in intensity. This limitation reflects the recorded data, necessitating a pragmatic approach. However, the different measures derived from the binary indicator may hint at intensity. Another limitation is combining loneliness and social isolation, though consistent with prior research [e.g. 12]. Distinguishing these concepts is important for targeted interventions; for instance, day centres reduce social isolation but may not address emotional loneliness [as conceptualised in e.g. 76]. While social care administrative records do not allow for such distinctions, insights from other datasets could better inform interventions [see e.g. 12,14,18–20,42,43].

Another notable limitation is related to the dataset which, although large in terms of sentence count, is limited to a relatively small geographic area. Although notes in the training and test set are not about the same individual, they may have been written by the same worker. Similarly, there may be organisational culture issues which lead to individuals using similar phrases that would not be seen elsewhere. We expect that the model will not perform quite as well on free text case notes from another area, although the magnitude of the dropoff, and how many new samples need to be labelled to improve performance, is an empirical question that we hope to answer in the future.

Studies using administrative data face inherent limitations when measuring phenomena like loneliness or isolation. While Table 7 shows that workers who record loneliness or isolation are more likely to arrange social inclusion services, it is unclear how often services are provided to individuals who are actually lonely or isolated. Some loneliness or isolation may remain unrecorded, and services may also be declined. However, the observed associations with characteristics like impaired memory and living alone suggest unobserved cases are not significant enough to invalidate our results. Administrative data provides real-time information on service use and has been used to link care home admission to factors like age, gender,

and disability [e.g. 77,78]. Including loneliness or isolation as a structured indicator in such models could further enhance such models.

## Conclusion

Our best-performing model achieves an $F_1$ score of 0.92 on unseen test data, demonstrating its accuracy for identifying loneliness and social isolation in long-term care case notes. The measure of loneliness and isolation seems valid, as it aligns with expected associations, such as living alone and impaired memory, and strongly predicts the receipt of social inclusion services. Approximately 43% of individuals had an assessment sentence indicating loneliness or isolation, 44% had a case note, 62% had either, and 26% had both. These prevalence estimates are comparable to survey data but benefit from administrative data's larger sample size of statutory care users, inclusion of high-needs individuals, and availability of time-variant service cost data, enabling detailed subgroup analyses and associations with service use.

Future research could use predictive outputs from administrative free text in regression models to explore variations in long-term care usage, such as the risk of care home entry. Our model enables such analyses, and highlights methods for extracting other characteristics not captured in structured data, such as economic hardship or psychological wellbeing. We provide an open-source version of the model in S1 Text, offering a foundation for researchers to apply it to their own data.

## Supporting information

**S1 Fig**. **Data flow diagram.** Diagram of data sharing agreements between data controller and data processors.
(TIF)

**S2 Fig**. **Data pre-processing.** Pre-processing steps taken with the count-based, skip-gram, and transformers vectors.
(TIF)

**S1 Text**. **Supplementary appendices and results.** This file contains:
  (1) Data Flow Appendix—Explanation of the pseudonymisation process, information governance, and data flow arrangements.
  (2) Methods Appendix—Details about the methods, including data pre-processing, labelling data, sentence vectors, model parameters, and classification rules.
  (3) Open-source model repository—Link to the model at GitHub.
  (4) Additional results—Tabular results corresponding to Fig 4.
(PDF)

## Acknowledgments

The authors extend their heartfelt gratitude to Uche Osuagwu for his tireless dedication to managing data extraction and quality, ensuring the data met the highest standards for academic research. We are deeply appreciative of William Wood and the Intelligence Solutions for London team for their vital contributions to Information Governance. Our sincere thanks go to Hannah Kendrick for her extraordinary generosity in dedicating her time and effort to establish Inter-Rater Reliability of human annotators. We are also immensely grateful to Explosion AI for providing us with a free research license for their proprietary annotation software, Prodigy.

## Author contributions

**Conceptualization:** Sam Rickman, Juliette Malley.

**Data curation:** Sam Rickman.

**Formal analysis:** Sam Rickman.

**Funding acquisition:** Sam Rickman, Jose-Luis Fernandez.

**Methodology:** Sam Rickman, Jose-Luis Fernandez, Juliette Malley.

**Project administration:** Jose-Luis Fernandez.

**Software:** Sam Rickman.

**Supervision:** Jose-Luis Fernandez, Juliette Malley.

**Visualization:** Sam Rickman.

**Writing – original draft:** Sam Rickman.

**Writing – review & editing:** Sam Rickman, Jose-Luis Fernandez, Juliette Malley.

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
