## [Decision Letter · Decision Letter 0]

14 Oct 2024

PONE-D-24-30453Using natural language processing to understand loneliness in English older long-term care users from free text case notesPLOS ONE

Dear Dr. Rickman,

Thank you for submitting your manuscript to PLOS ONE. After careful consideration, we feel that it has merit but does not fully meet PLOS ONE’s publication criteria as it currently stands. Therefore, we invite you to submit a revised version of the manuscript that addresses the points raised during the review process.

I have read the reviewer's comments, and I believe they are valid and will improve the manuscript.

I have a question regarding the performance of traditional machine learning algorithms, particularly logistic regression. If you use scikit-learn, I wonder if the parameter class_weight is set to "balanced" instead of the default.

We look forward to receiving your revised manuscript.

Kind regards,

Mario Graff-Guerrero, Ph.D.

Academic Editor

PLOS ONE

“This paper is based on independent research funded through the NIHR Policy Research Unit in Adult Social Care, reference PR-PRU-1217-21101. Funding was also received from the UK National Institute of Health and Care Research (NIHR) Applied Research Collaboration (ARC) North Thames under grant number NIHR200163. An additional grant was received from the NHS Digital Social Care Pathfinders initiative under the contract 8717. The views expressed are those of the authors and not necessarily those of the NIHR, ARC, NHS, or the Department of Health and Social Care.”

“The authors extend their heartfelt gratitude to Uche Osuagwu for his tireless dedication

to managing data extraction and quality, ensuring the data met the highest standards

for academic research.

We are deeply appreciative of William Wood and the Intelligence Solutions for London

team for their vital contributions to Information Governance.

Our sincere thanks go to Hannah Kendrick for her extraordinary generosity in

dedicating her time and effort to establish Inter-Rater Reliability of human annotators.

We are also immensely grateful to Explosion AI for providing us with a free research

license for their proprietary annotation software, Prodigy.

This paper is based on independent research funded through the NIHR Policy Research

Unit in Adult Social Care, reference PR-PRU-1217-21101. Funding was also received

from the UK National Institute of Health and Care Research (NIHR) Applied Research

Collaboration (ARC) North Thames under grant number NIHR200163. An additional

grant was received from the NHS Digital Social Care Pathfinders initiative under the

contract 8717. The views expressed are those of the authors and not necessarily those of

the NIHR, ARC, NHS, or the Department of Health and Social Care.”

“This paper is based on independent research funded through the NIHR Policy Research Unit in Adult Social Care, reference PR-PRU-1217-21101. Funding was also received from the UK National Institute of Health and Care Research (NIHR) Applied Research Collaboration (ARC) North Thames under grant number NIHR200163. An additional grant was received from the NHS Digital Social Care Pathfinders initiative under the contract 8717. The views expressed are those of the authors and not necessarily those of the NIHR, ARC, NHS, or the Department of Health and Social Care.”

5. For studies involving third-party data, we encourage authors to share any data specific to their analyses that they can legally distribute. PLOS recognizes, however, that authors may be using third-party data they do not have the rights to share. When third-party data cannot be publicly shared, authors must provide all information necessary for interested researchers to apply to gain access to the data. (https://journals.plos.org/plosone/s/data-availability#loc-acceptable-data-access-restrictions)

a) A description of the data set and the third-party source

b) If applicable, verification of permission to use the data set

c) Confirmation of whether the authors received any special privileges in accessing the data that other researchers would not have

d) All necessary contact information others would need to apply to gain access to the data

Reviewers' comments:

Reviewer's Responses to Questions

**Comments to the Author**

1. Is the manuscript technically sound, and do the data support the conclusions?

Reviewer #1:Yes

Reviewer #2:Yes

2. Has the statistical analysis been performed appropriately and rigorously? 

Reviewer #1:Yes

Reviewer #2:Yes

3. Have the authors made all data underlying the findings in their manuscript fully available?

Reviewer #1:No

Reviewer #2:No

4. Is the manuscript presented in an intelligible fashion and written in standard English?

Reviewer #1:Yes

Reviewer #2:Yes

5. Review Comments to the Author

Reviewer #1:This manuscript presents the development of NLP methods to extract information on loneliness/social isolation from clinical notes of older adults in a London borough. Additionally, it correlates these findings with established social isolation scales such as CES, SNI, UCLA, and other data. The study addresses a relevant issue, especially in the context of loneliness during and after the COVID-19 pandemic. However, the reviewer has several major concerns that need to be addressed.

Major Comments:

1. The manuscript covers a broad range of topics, from NLP method development to comparisons with multiple outcomes and activities. The current title does not reflect the full scope of the work. The authors may want to consider splitting the study into separate publications—one focused on NLP methodology and another on the comparative analysis of loneliness/social isolation with various outcomes.

2. The manuscript reports that 3,046 patients have 1.1 million notes, which seems unusually high. It would be helpful to provide more details on the types of notes, their categories, and the nature of the visits associated with them.

3. The authors mention correlating loneliness/social isolation with 'living alone,' but it is unclear how this was identified from the clinical notes. More details on the methodology used to determine this aspect would be beneficial.

4. The manuscript lacks a clear definition of 'loneliness' and 'social isolation.' It is important to note that these terms are not synonymous. One of the recent work (https://arxiv.org/pdf/2403.17199) shows distinctions between these concepts. Including definitions and illustrative examples from the notes would improve the manuscript clarity.

4. The following key studies on social isolation and adverse social support should be cited:

https://arxiv.org/pdf/2403.17199

https://www.nature.com/articles/s41746-023-00970-0

5. The manuscript lacks comparison with existing studies on loneliness and social isolation in clinical settings. It would also be useful to see baseline methods using rule-based approaches for comparison with the NLP methods.

6. The manuscript does not provide sufficient information on how CES, SNI, and UCLA data were collected. Clarifying this will be crucial for understanding the validity of the correlations made in the study.

7. Table 8 presents interesting observations, but the manuscript does not explain how the underlying data were collected.

Minor Comments:

1. Use "1.1 million" instead of "1.1m".

2. Table 8 is referred to far earlier in the text than it is presented.

3. Consider merging Tables 5 and 6 to streamline the presentation of the data.

Reviewer #2:The paper applies NLP techniques to create an index for loneliness or social isolation. Their best model is based on Roberta, and it scored 0.92 using F1. The paper then develops a series of tests to compare the model to survey data.

Since the importance of the study is evident, I have some comments that could benefit the final manuscript.

Comments

It will help the readability if you include some aspects of the appendices in the text, for example, how to pseudonymize the data or the labeling process. Also include the rationale for some of the decisions like replacing masks with "real text".

Explain the content of your data in the "Data" section. Later in the text, you mention it includes gender, ethnicity, age, mobility, demographic, etc. You can describe your data here.

Line 279 "In order to establish that our best-performing model was not picking up demographic characteristics, or simply those cases where there are more notes, we also included these in the model" Here, the model refers to the logistic regression model right? Then, it is not clear how adding some characteristics to the LR model establishes that the Roberta model was not pickin' up those.

The manuscript could benefit from a significant reordering of the text. The section "Model Evaluation" can be explained in the Results section together with the tables from the experiments. You could also include there the discussion of the findings. This could simplify the reading and the flow of the paper. For example, the sentences on line 404: "This difference may not explain the significant association in ELSA between unpaid care and loneliness or isolation" is referring to table 5 (I guess) which appeared 4 pages before.

Figure 4 is unreadable.

line 327 "The χ2 results indicate that living alone and shopping are the only indicators associated in ELSA with both measures of loneliness and social isolation" So if we are comparing CES and UCLA columns with the Both columns, what about Contincence and Meal prep?

In apendix, the sections Classification rules and Lonely or socially isolated are empty.

In the appendix, page 5: "Notes from the test set which contained these terms were then manually classified by human annotators." How were the notes in the train set annotated?

Minor Issues

Introduction

line 2 public expenditure [was] on long-term care was 1.98%

The impact of loneliness and social isolation

line 62 first appearence of term ELSA without a definition (which is on line 65)

Materials and methods

Data

line 94 In Fig 1 we show how [the] the assessment

Data access and time frame

line 97 Write the meaning of NHS

Descriptive statistics

Methods overview

Model development

Model Evaluation

line 241 Explain what are the waves 6-9 of ELSA.

line 263 individuals in the [the]administrative

Equation 1 The order of the variables and their explanation in the text don't match.

Results

line 338 Define GVIF.

Table 6 The columns ELSA and Administrative Records are in reversed order according to Table 5.

Table 6 What is the difference between CES-D and CES?

Table 6: The legend for the * and . marks is wrong.

Discussion

line 390 it may be the case [that]that the individuals

Limitations

line 490 The paragraph appeared on line 475.

line 505 Table 7.[ .]

6. PLOS authors have the option to publish the peer review history of their article (what does this mean?). If published, this will include your full peer review and any attached files.

Reviewer #1:No

Reviewer #2:No

While revising your submission, please upload your figure files to the Preflight Analysis and Conversion Engine (PACE) digital diagnostic tool,https://pacev2.apexcovantage.com/. PACE helps ensure that figures meet PLOS requirements. To use PACE, you must first register as a user. Registration is free. Then, login and navigate to the UPLOAD tab, where you will find detailed instructions on how to use the tool. If you encounter any issues or have any questions when using PACE, please email PLOS atfigures@plos.org. Please note that Supporting Information files do not need this step.

---

## [Author Response · Author response to Decision Letter 1]

25 Oct 2024

We are very grateful to the editor and reviewers for their time and insightful comments. We have now refined the manuscript based on these remarks. We include the point-by-point responses in the Response to Reviewers document.

---

## [Decision Letter · Decision Letter 1]

6 Dec 2024

PONE-D-24-30453R1Understanding patterns of loneliness in older long-term care users using natural language processing with free text case notesPLOS ONE

Dear Dr. Rickman,

Thank you for submitting your manuscript to PLOS ONE. After careful consideration, we feel that it has merit but does not fully meet PLOS ONE’s publication criteria as it currently stands. Therefore, we invite you to submit a revised version of the manuscript that addresses the points raised during the review process.The manuscript is going in the right direction, one of the reviewers accept the current version, and the otherhas some concerns and madesuggestions to improve the manuscript, please read them carefully. In my opinion, the comments contribute to improving the paper.

We look forward to receiving your revised manuscript.

Kind regards,

Mario Graff-Guerrero, Ph.D.

Academic Editor

PLOS ONE

Journal Requirements:

Reviewers' comments:

Reviewer's Responses to Questions

**Comments to the Author**

1. If the authors have adequately addressed your comments raised in a previous round of review and you feel that this manuscript is now acceptable for publication, you may indicate that here to bypass the “Comments to the Author” section, enter your conflict of interest statement in the “Confidential to Editor” section, and submit your "Accept" recommendation.

Reviewer #1:All comments have been addressed

Reviewer #2:All comments have been addressed

2. Is the manuscript technically sound, and do the data support the conclusions?

Reviewer #1:Partly

Reviewer #2:Yes

3. Has the statistical analysis been performed appropriately and rigorously? 

Reviewer #1:Yes

Reviewer #2:Yes

4. Have the authors made all data underlying the findings in their manuscript fully available?

Reviewer #1:No

Reviewer #2:No

5. Is the manuscript presented in an intelligible fashion and written in standard English?

Reviewer #1:Yes

Reviewer #2:Yes

6. Review Comments to the Author

Reviewer #1:The authors developed NLP systems to identify social isolation and loneliness. They compared the NLP-derived output with other collected data, such as ELSA, SNI, and UCLA loneliness scores. Below are my comments:

1. The abstract and manuscript can be significantly condensed without losing essential details.

2. Recent literature uses "SDOH" as a short form for Social Determinants of Health. The authors are encouraged to adopt similar abbreviations for consistency.

3. All figure and table captions are too brief. These should be expanded for clarity. Also, in Table 1, replace "N notes" with "# Notes" and "N words" with "# Words."

4. The numbers in Table 2 do not align with the 10,083 training sentences and 3,573 test sentences. If these are not identical, the authors should clarify how they differ and provide a detailed explanation.

5. The manuscript mentions a feed-forward neural network alongside RoBERTa and DistilRoBERTa. These are distinct architectures and should not be conflated.

6. The equations for precision, recall, and F-score can be moved to the supplementary materials to streamline the manuscript.

7. To improve clarity, the construct validity results could be presented in a table format within the results section.

8. It would enhance readability if all figures were included within the main text rather than at the end of the publication.

9. The discussion section should be divided into multiple subsections aligned with the results for better organization and coherence.

10. The authors should address why some categories exhibit high accuracy but have zero precision, recall, and F-scores.

11. Including counts of loneliness/social isolation versus no loneliness/no social isolation in both the training and test datasets would be valuable for understanding data balance.

12. It would be helpful if the authors could provide a detailed review response letter outlining the changes made in the text.

Reviewer #2:All my comments were attended to. I have no further recommendations. Since your data is anonymized, you could make it public.

7. PLOS authors have the option to publish the peer review history of their article (what does this mean?). If published, this will include your full peer review and any attached files.

Reviewer #1:No

Reviewer #2:No

While revising your submission, please upload your figure files to the Preflight Analysis and Conversion Engine (PACE) digital diagnostic tool,https://pacev2.apexcovantage.com/. PACE helps ensure that figures meet PLOS requirements. To use PACE, you must first register as a user. Registration is free. Then, login and navigate to the UPLOAD tab, where you will find detailed instructions on how to use the tool. If you encounter any issues or have any questions when using PACE, please email PLOS atfigures@plos.org. Please note that Supporting Information files do not need this step.

---

## [Editor Report · Decision Letter 2]

7 Feb 2025

Understanding patterns of loneliness in older long-term care users using natural language processing with free text case notes

PONE-D-24-30453R2

Dear Dr. Rickman,

We’re pleased to inform you that your manuscript has been judged scientifically suitable for publication and will be formally accepted for publication once it meets all outstanding technical requirements.

An invoice will be generated when your article is formally accepted. Please note, if your institution has a publishing partnership with PLOS and your article meets the relevant criteria, all or part of your publication costs will be covered. Please make sure your user information is up-to-date by logging into Editorial Manager at Editorial Manager®and clicking the ‘Update My Information' link at the top of the page. If you have any questions relating to publication charges, please contact our Author Billing department directly at authorbilling@plos.org.

Kind regards,

Mario Graff-Guerrero, Ph.D.

Academic Editor

PLOS ONE

Additional Editor Comments (optional):

I enjoyed reading the manuscript. Congratulations on the excellent research!
---

## [Editor Report · Acceptance letter]

PONE-D-24-30453R2

PLOS ONE

Dear Dr. Rickman,

I'm pleased to inform you that your manuscript has been deemed suitable for publication in PLOS ONE. Congratulations! Your manuscript is now being handed over to our production team.

Kind regards,

on behalf of

Dr. Mario Graff-Guerrero

Academic Editor

PLOS ONE